# The Bio-Safety Concerns of Three Domestic Temporary Hair Dye Molecules: Fuchsin Basic, Victoria Blue B and Basic Red 2

**DOI:** 10.3390/molecules24091744

**Published:** 2019-05-05

**Authors:** Bing Liu, Shu-Fang Jin, Hua-Chao Li, Xiang-Yu Sun, Si-Qi Yan, Shu-Jun Deng, Ping Zhao

**Affiliations:** 1Guangzhou Key Laboratory of Construction and Application of New Drug Screening Model Systems, Guangdong Pharmaceutical University, Guangzhou 510006, China; liubing520@gdpu.edu.cn (B.L.); rehclee27@163.com (H.-C.L.); 2Key Laboratory of New Drug Discovery and Evaluation of Ordinary Universities of Guangdong Province, Guangdong Pharmaceutical University, Guangzhou 510006, China; 3School of Chemistry and Chemical Engineering, Guangdong Pharmaceutical University, No. 280, Waihuandong Road, Education Mega Centre, Guangzhou 510006, China; jinshufang333@126.com (S.-F.J.); 18500391080@163.com (X.-Y.S.); yimxikei@163.com (S.-Q.Y.); 18344005295@163.com (S.-J.D.)

**Keywords:** hair dyes, hemolytic effect, cytotoxic effects, BSA, DNA

## Abstract

Hair-coloring products include permanent, semi-permanent and temporary dyes that vary by chemical formulation and are distinguished mainly by how long they last. Domestic temporary hair dyes, such as fuchsin basic, basic red 2 and Victoria blue B, are especially popular because of their cheapness and facile applications. Despite numerous studies on the relationship between permanent hair dyes and disease, there are few studies addressing whether these domestic temporary hair dyes are associated with an increased cancer risk. Herein, to ascertain the bio-safety of these temporary hair dyes, we comparatively studied their percutaneous absorption, hemolytic effect and cytotoxic effects in this paper. Furthermore, to better understand the risk of these dyes after penetrating the skin, experimental and theoretical studies were carried out examining the interactions between the dyes and serum albumins as well as calf thymus (CT)-DNA. The results showed that these domestic temporary hair dyes are cytotoxic with regard to human red blood cells and NIH/3T3 cell lines, due to intense interactions with bovine serum albumin (BSA)/DNA. We conclude that the temporary hair dyes may have risk to human health, and those who use them should be aware of their potential toxic effects.

## 1. Introduction

Given their widespread use, the impact of hair dyes on human health has been a topic of intensive study and discussion in recent years [1,2]. Hair-coloring dyes can be categorized into permanent, semi-permanent, and temporary dyes that vary by chemical formulation and their persistence on the hair. Permanent dye materials have been studied substantially in recent decades and it has been suggested that they are closely related to an increased risk of cancer, such as bladder, lung, ovarian and non-Hodgkin’s lymphoma [3,4,5]. For instance, p-Phenylenediamine (PPD), which was used in more than 80% of permanent dyes, is frequently reported to cause vascular neuroedema, acute renal failure or bladder cancer [6,7]. In the last five years, many permanent and semi-permanent hair dyes have been banned in order to guarantee consumers’ safety [8].

Despite significant research on permanent hair dye materials, the study on the relationship between cancer and temporary hair dyes is still in its infancy [9]. Especially owing to their inexpensive formulations and easy application, temporary hair dyes permit domestic usage and have been very popular in recent years. Some temporary dye molecules, such as fuchsin basic (FB), basic red 2 (BR2) and Victoria blue B (VBB), are widely accepted as “safe” dye materials that simply deposit on the surface of the hair without penetrating into the cortex [10,11]. Moreover, an epidemiology review also identified that use of temporary hair dyes appears to have little evidence of an association with cancer or other adverse events [12].

Recently, some toxicologists have challenged this viewpoint and begun to shine the light on the toxicity of these dye materials, since after cleavage many of the temporary hair dye molecules can also form toxic aromatic amines like PPD [13]. Orange 1, which is frequently employed in temporary hair dyes, was reported to induce cell death by necrosis in mouse fibroblasts [14]. Other temporary hair dyes, such as basic red 51, were also indicated to have anticancer toxic potential in cell experiments [15]. Thus, whether temporary hair dyes are associated with an increased cancer risk has been controversial and there is limited available research addressing this issue [16].

Herein, we examined the bio-safety of three domestic temporary hair dyes: Fuchsin basic (FB), Victoria blue B (VBB) and basic red 2 (BR2, see molecular structures in Figure 1). Percutaneous absorption, hemolytic effect and cytotoxic effects to NIH/3T3 cell lines of these dyes were studied under the permitted dosage ratios in this report. The NIH/3T3 cell belongs to the mouse embryonic fibroblast cell line and is commonly employed in the research of drug cytotoxicity to human skins [17,18]. Furthermore, the interactions between the dye molecules and serum albumins as well as calf thymus (CT)-DNA were also investigated in depth. Our aim in this work was to assess the risks and reveal the toxic mechanism of the three temporary hair dyes at the cellular and molecular levels.

## 2. Results

### 2.1. Percutaneous Absorption

The percutaneous absorption measurements for FB, BR2 and VBB dyes were carried out by following a standard protocol with diffusion cells. The results are given in Figure 2, where the instrument of this experiment was inserted.

From Figure 2, the potential absorption of BR2 through the skin had a penetrating concentration up to 23 μM after 48 h. The penetrating concentrations of these dyes gradually rose as time extended. The BR2 exhibited much higher percutaneous absorption ability than FB and VBB, with or without oxidant. This indicated that the BR2 was not an ideal candidate as a hair dye in this respect. VBB had relatively less percutaneous absorption than the other two dyes, under both experimental conditions.

### 2.2. Hemolytic and Bacteria Cytotoxic Effects of Hair Dyes

To assess the toxicity of these dyes on human erythrocytes, hemolytic assays of FB, BR2 and VBB were performed. Table 1 illustrates the LD50 (lethal dose) values to the rupture of the human red blood cells (RBC) of these hair dyes, and the RBC hemolytic details of these dyes are given as Figure 3. From Figure 3, it was noted that the hemolytic rate of RBC increased significantly with the addition of the dye concentrations, demonstrating a dose response. Moreover, Table 1 indicates that VBB exhibited a much higher hemolytic efficiency than FB and BR2, although it had the advantage of low percutaneous absorption suggested by the experiment above.

### 2.3. Cell Cytotoxicity of Dyes

As the concentrations of the hair dyes present in the solution increased, the cell viability significantly decreased, demonstrating dose response on the cell systems (Figure 4). The lethal dose 50% (LD50) values of the tetrazolium dye (MTT) assay for NIH/3T3 cells varied for different dye systems and were statistically significant (Table 1)

### 2.4. BSA Interactions of Dyes

#### 2.4.1. BSA Absorption Change with Dyes

It is well-established that formation of a complex between protein and substance could increase or decrease the UV-vis spectrum of protein. Figure 5 shows the UV-vis absorption spectra of BSA in the absence and presence of VBB, BR2 and FB. It was found that the absorbance of BSA at 280 nm, which could be contributed by the amino acids phenylalanine (Phe), tyrosine (Tyr), and tryptophan (Trp), changed remarkably after the addition of the dyes [19,20]. The change in the peak intensity at 280 nm demonstrated that the binding of dyes to BSA could lead to the loosening and unfolding of BSA conformation while increasing the hydrophobicity of the micro-environment of the aromatic amino acid residues [21]. These results indicated that these dyes had interactions with the protein and may present risks when used as hair dyes.

#### 2.4.2. Dyes’ Fluorescence Emission Change with BSA

Figure 6 gives the fluorescence emission spectra of BSA with the titration of FB as an example. The emission intensity was quenched significantly with the addition of FB. Similar results were obtained for VBB and BR2 (Appendix A). It is widely accepted this fluorescence quenching is the direct evidence for the binding of substrates to proteins [22,23].

It is well known that quenching could occur through the static or dynamic quenching process. The Stern–Volmer equation (Equation (1)) is often employed to analyze the quenching mechanism [23,24]:(1)$$\frac{F_{0}}{F_{}}=1+Kq\zeta_{0}[Q]=1+Ksv[Q]$$ where F and F_0_ are the fluorescence intensities in the presence and absence of a quencher, respectively. Ksv, Kq, and Kb [Q] denote the Stern–Volmer constant, quenching rate constant, the original lifetime of protein, and the concentration of quencher, respectively.

To better understand the interaction mechanism between the dye and BSA, Table 1 exhibits the Kq of the studied dyes, from which we found that Kq were in a range of 1.10 × 10^11^ to 2.34 × 10^12^, which is much larger than the largest dynamic quenching constant for biological molecules, 2.0 × 10^10^ [3].Therefore, we suppose that it is the static rather the dynamic quenching mechanism resulting in the fluorescence quenching of dyes. Meanwhile, from Table 1, the Kq for FB was higher than those for VBB and BR2, suggesting that the quenching ability of FB for BSA was larger than the other two dyes.

Equation (1) is also frequently employed to obtain the binding constant Kb and binding site number *n* between BSA and the dyes, which assumes the existence of *n* independent binding sites for ligands on the protein molecule. Equation (1) is often expressed as Equation (2) in calculating Kb and *n*:(2)$$lg[\frac{F_{0}-F}{F}]=lgK_{b}+nlg[Q]$$

The BSA-binding parameters of the studied dyes calculated using Equation (2) are also listed in Table 1. The binding constant Kb of FB was much larger than VBB and BR2. The binding site numbers (*n*) of all the dyes on BSA were about 1, indicating that BSA had only one site for the binding of these dyes.

#### 2.4.3. The Molecular Docking of the Dyes with BSA

The UV-vis and the fluorescence spectroscopy results were complemented with the computational molecular docking study. Figure 7 presents the most stable conformations between BSA and the dyes based on the lowest free binding energy while details of the binding hydrogen bonds for the dye–BSA complex are summarized in Table 2. The details of the H-bonds between the BSA and dyes are given in Appendix A in the Appendix A. It is understandable that various hydrophobic and hydrophilic amino acids were in contact with the dyes in dye–BSA complex. For FB, H-bonds are at a distance of 2.88 Å between N1 of FB and NE2 of His 145. As to VBB, H-bonds exist between N2, N3 and N4 of the VBB molecule with NZ of Lys499, OE2 of Glu125, O of Leu115, respectively.

Meanwhile, hydrophobic bonds also existed between these dyes and BSA. FB was surrounded by Arg144, Tyr 137, Glu140, Leu115, Tyr160, Leu122, Pro117, Ile181, Arg185, Glu182, Ile141, Leu189, Met184, and Lys136 amino acid residues with the lowest binding energy of −7.66 kcal/mol (see Figure 7a). VBB was surrounded by Val481, Leu346, Trp213, Leu480, Ala209, Ala349, Leu330, Glu353, Val215, Gly327, Asp323, Lys350, Asg208, Ala212, Tyr137, Tyr160, Leu122, Leu115, Leu189, Arg144, His145, Val188, Ile141, Arg185, Pro117, Leu178, Glu182, and Ile1181 amino acid residues with the lowest binding energy of −9.9 kcal/mol (see Figure 7b). BR2 was surrounded by Pro415, Tyr496, Lys533, Glu530, Ser418, Thr419, Val468, Val417, Thr466 Phe133, Tyr137, Lys136, Tyr160, Pro117, Leu122, Asp118, Lys116, and Arg185 amino acid residues with the lowest binding energy of −7.85 kcal/mol (see Figure 7c). Collectively, these molecular docking studies give a vivid illustration for the intense interactions between BSA and these dyes, which were evidenced by the spectral experiments.

### 2.5. DNA Interactions of Dyes

#### 2.5.1. DNA Absorption Change with Dyes

The binding of small molecules to DNA produces hypochromism, a broadening of the envelope, and a red shift of the complex absorption band [25]. Absorption titration experiments were employed to examine the association of the dyes with CT-DNA, and the spectral changes are exhibited in Figure 8. The addition of CT-DNA to the solutions of FB, BR2 and VBB resulted in a significant decrease in absorption followed by a slight red shift, indicating a shift to a more polar environment on interaction. The large spectral changes of the dyes with the addition of DNA suggested the intense interactions between CT-DNA and the dyes.

To quantitatively compare the affinities of the dyes bound to DNA, the intrinsic binding constants Kb were measured and the results are given in Table 1. The DNA binding constants for the studied dyes follow an order of BR2 > VBB >> FB. Cationic dyes (BR2 and VBB) exhibited much higher DNA binding affinities than the neutral dye (FB), which may be easily understood by the negative DNA phosphate backbone. Static electronic strength plays the key role in the binding of BR2 and VBB with DNA. FB may mainly employ the hydrophobic strength in the DNA interaction.

#### 2.5.2. Dyes’ Fluorescence Emission Change with CT-DNA

The interactions between the three hair dyes and CT-DNA were also probed using fluorescence techniques. Emission spectra for the dye ligands were recorded in DNA free solutions and in the presence of varying amounts of CT-DNA. The fluorescence emission spectra changes observed for FB, BR2 and VBB are shown in Figure 9. With the increase of CT-DNA, the fluorescence emissions of these dyes were significantly quenched. This could be explained similarly to the case of BSA, which is an indication of interaction between dyes and DNA [26].

#### 2.5.3. The Molecular Docking of the Dyes with CT-DNA

In order to determine the binding details of these dyes with CT-DNA, molecular docking experiment was performed. Figure 10 exhibits the free molecular binding energy (M.B.E.) of the dyes with DNA, from which it can clearly be seen that the BR2 had lower binding energies with DNA than other dyes. This indicated that the binding interaction between BR2 and DNA scaffold was more stable than that between the other two dyes with DNA. This result well interpreted the DNA binding behaviors and affinities above.

The details of the most stable conformations between the dyes and DNA structures were described based on the lowest free binding energy, and the results are given in Figure 10. From Figure 10, a major finding was that BR2 could intercalate into the DNA bases while the other two dyes bound at the grooves of the DNA skeletons. Intercalation was the most favorable binding model in the ligand–DNA binding and thus the BR2 had the lowest binding energy and highest DNA affinity.

Hydrophobic interaction also played a significant role in the dye–DNA binding. Table 2 exhibits the hydrogen bonds for dye–CT-DNA complexes and Appendix A in the Appendix A gives the surrounding details of these dyes. For FB, the N1, N2 formed H-bonds with N2 of dg10, O4 of dc, and O2 of dt8, respectively. Meanwhile, H-bonds also existed in BR2 and VBB with DNA structures. BR2 formed H-bonds with O4 of Dt19 at a distance of 3.06 Å while for VBB, H-bonds were at a distance of 2.17 Å between N2 and O4 of Da6. The H-bonds in BR2-DNA and VBB-DNA were relatively less than in FB–DNA interaction. This could be explained by the fact that in the cationic molecules, electrostatic strength takes the majority in the binding behaviors. Thus, the theoretical results visually explained the DNA binding behaviors of these dyes.

## 3. Discussion

In recent years, a controversial issue over whether temporary hair dyes are associated with an increased cancer risk has aroused people’s attention. Herein, the risks of three permitted temporary hair dyes, fuchsin basic (FB), Victoria blue B (VBB) and basic red 2 were assessed under their permitted dosage ratios.

### 3.1. Percutaneous Absorption

Skin is the first and the main protection for both consumers and hairdressers using hair dyes. A number of hair dyes are of concern from a toxicologic standpoint because of their potential absorption through the exposed skin. Several hair dyes that are carcinogenic in animal feeding studies are known to be absorbed through human skin [27,28]. The percutaneous absorption results shown in Figure 2 illustrated that the penetrating concentrations of these dyes gradually rose as time extended. The BR2 exhibited much higher percutaneous absorption ability than FB and VBB, with or without oxidant.

The different percutaneous absorption of these dyes could best be understood by their different molecular structures. It is well known that the membrane of skin cells is lipoid, but the interstitial fluid of the cell is polar. Thus, the molecular structure of the dyes plays an important role in the percutaneous efficiency [29,30]. VBB has a more rigid structure with more benzene rings than FB and BR2. This rigid structure may result in an accumulation of VBB in the stratum corneum instead of the percutaneous absorption of the skin. In contrast, BR2, with a more symmetric structure, may have a proper oil/water partition coefficient, and thus could penetrate the skin more easily.

The percutaneous absorption experiment indicated that the studied dyes could penetrate the skin to different extents. It is well accepted that the toxicity of the hair dyes may result in the rupture of erythrocytes and the cytoplasmic contents in the erythrocytes may then release into the surrounding fluids [31,32]. VBB exhibited a much higher hemolytic efficiency that FB and BR2, although it had the advantage of low percutaneous absorption. VBB was favored to penetrate the cell membrane more easily than the other two dyes, which may also result from its lip-soluble molecular structure with more aromatic rings.

### 3.2. Effects to Human

Mouse embryonic fibroblast NIH/3T3 cell line is widely employed in the research of drug cytotoxicity to human skins. All these three hair dyes were cytotoxic to NIH/3T3 cell systems, indicating that the exposure of these dyes to human skin is harmful. FB exhibited the highest cytotoxicity compared to the other two hair dyes, suggesting that it is relatively more toxic when employed as a hair dye.

### 3.3. BSA and DNA Interactions of Dyes

Serum albumins are abundantly found in blood plasma and belong to one of the most widely studied categories [33,34,35]. They function as carriers for numerous exogenous and endogenous compounds in the body. On the other hand, DNA has been reported as an important target for the toxicity of many hair dyes [17,36,37], which lead us to research the interactions between the studied dyes and DNA. The toxic mechanism of the hair dyes was studied by investigating their interactions with BSA/DNA.

Absorption and fluorescence spectra were employed to study the interactions between BSA/DNA and the hair dyes. The absorption spectral results suggested that the binding of dyes to BSA could lead to the loosening and unfolding of BSA conformation while increasing the hydrophobicity of the micro-environment of the aromatic amino acid residues. Moreover, the Kq of the studied dyes were larger than the largest dynamic quenching constant for biological molecules, indicating that it was the static rather the dynamic quenching mechanism resulting in the fluorescence quenching of dyes. Thus, these dyes have interactions with the protein and present risk when used as hair dyes.

The spectral results also indicated that all the studied dyes have tight interaction with DNA. Cationic dyes (BR2 and VBB) exhibited much higher DNA binding affinities than the neutral dye (FB), which could be easily understood by the negative DNA phosphate backbone. Static electronic strength plays the key role in the binding of BR2 and VBB with DNA. FB may mainly employ the hydrophobic strength in the DNA interaction. The molecular docking calculation further indicated that BR2 could intercalate into the DNA bases while the other two dyes bound at the grooves of the DNA skeletons. Intercalation was the most favorable binding model in the ligand–DNA binding and thus the BR2 has the lowest binding energy and highest DNA affinity. The three dyes were confirmed to have intense interaction with DNA, which may further affect the DNA replication. Further research on the toxic mechanism for these hair dyes is currently undergoing clinic trails.

## 4. Materials and Methods

### 4.1. Materials and Instruments

All chemicals used within this experiment were of analytical quality and used without any further purification.

Calf thymus DNA (CT-DNA) and BSA were purchased from the Shanghai Sangon Biological Engineering Technology & Services. Co., Ltd. (Shanghai, China). All dyes were purchased from Beijing Coupling Technology. Co., Ltd. (Beijing, China). The People’s Hospital of Zhongshan provided the fresh human blood samples. PBS buffer consisted of 10 mM K_2_HPO_4_/KH_2_PO_4_, 40 mM KCl, 100 mM NaCl, pH 7.4. Pig abdominal skin was lightly shaved with electric clippers in vitro experiments, using care to prevent damage to the skin. The absorption through pig abdominal skin was measured by in vitro diffusion cell techniques. UV-Vis spectra were recorded on a Hitachi U-3900H spectrophotometer (Hitachi, Tokyo, Japan). Fluorescence spectral experiments were recorded on a PerkinElmer L55 spectrofluorophotometer (Perkin–Elmer, Norwalk, CT, USA) at room temperature. Cell viability assay was performed with a microplate reader (model 680, Bio-Rad, Hercules, CA, USA).

### 4.2. Methods

#### 4.2.1. Percutaneous Absorption

The absorption through excised pig abdominal skin was measured by in vitro diffusion cell techniques. A newly designed flow-through cell was used to obtain continuous monitoring for absorption profiles (Figure 2, insert); normal saline was pumped through the cells (skin surface area, 0.64 cm^2^) at a rate of approximately 5 mL/h and collected in scintillation vials for counting. Dermatome sections of pigskin were utilized in the permeability studies, lightly shaved with electric clippers as in vitro experiments, using care to prevent damage to the skin. The upper 300 μm of skin was removed from the skin surface. The volume of hair dyes solution was 25 mL. The solution was supplemented when the volume was less than 15 mL. FB and VBB dissolved in pure water and mixed with H_2_O_2_ were applied to skin in at the largest allowed dosage ratios of Hygienic Standard for Cosmetics, with 0.3% (the largest amount) and 0.15% (when mixed with oxidants) for FB, 0.5% (the largest amount) and 0.25% (when mixed with oxidants) for VBB, and 0.4% (the largest amount) and 0.2% (when mixed with oxidants) for BR2, respectively. The site of application was washed with soap and water at 24 h or 48 h in all experiments. Absorbed dyes were determined by the UV instrument.

#### 4.2.2. Hemolytic Effect of Hair Dye

To check the hemolytic effect of dye, blood samples of healthy individuals were collected from the pathology center in a tube containing an anticoagulant. Plasma and serum were removed from the sample by centrifuging at 5000 rpm for 5 min. Then, different concentrations of dyes for 0.05, 0.1, 0.15, 0.2, 0.25, and 0.3 mM were mixed with the RBC sample (1 mL of 10% solution in PBS, pH 7.4) and incubated for 30 min at 37 °C. A 0.1% Triton X-100 was considered the positive control. After completion of the incubation period, cell samples were centrifuged at 5000 rpm for 10 min and the absorbance of the supernatant containing lysed erythrocytes was measured at 540 nm. The percentage hemolysis of RBC was determined by the following equation:%Hemolysis = [A_t_ − A_c_/A_100%_ − A_c_] × 100(3) where A_t_ is the absorbance of the supernatant from samples treated with the dye; A_c_ is the absorbance of the supernatant from control (PBS); and A_100%_ is the absorbance of the supernatant from positive controls incubated in the presence of 0.1%Triton X-100, which causes complete lysis of RBC.

#### 4.2.3. Measurement of Cell Viability

For cytotoxicity assays, NIH/3T3 cells, originally obtained from American Type Culture Collection (ATCC), were seeded for 24 h in standard 96-well plates at 6 × 10^3^ cells per well. The culture medium was then discarded and the cells were treated for 48 h with 200 μL of medium containing different dye concentrations (6–100 μM). Cell viability was determined using a tetrazolium dye (MTT) assay. The cells were rinsed thrice with PBS pH 7.4 and incubated for 4 h in 100 μL of medium containing 0.5 g/L of MTT. Then the medium was replaced by 150 μL of dimethylsulfoxide to dissolve the formazan crystals formed by viable cells. Absorbance was measured at 490 nm using a microplate reader. The 50% inhibitory concentration (IC50) was determined as the dyes’ concentration that resulted in a 50% reduction in cell viability. All the experiments were performed in quintuplicate.

#### 4.2.4. Spectral Experiments

The absorption and fluorescence titrations were carried out by the stepwise addition. 10 mM DNA/BSA prepared in Tris buffer solution was added stepwise to the sample cell. After equilibration for 5 min, absorption or fluorescence spectra were recorded. The titration processes were repeated until there was no spectral change for at least three titrations, indicating the binding saturation had been achieved.

For absorption experiment, to compare quantitatively the affinities of the dyes bound to DNA, the intrinsic binding constants K’_b_ were measured by monitoring the changes of absorbance with increasing concentration of DNA using the following equation:[DNA]/(ε_a_ − ε_f_) = [DNA]/(ε_b_ − ε_f_) = +1/{K’b (ε_b_ − ε_f_)}(4) where [DNA] is the concentration of DNA in base pairs, ε_a_, ε_f_ and ε_b_ correspond to the apparent absorption coefficient A_obsd_/[Dye] (A_obsd_ refers to the observed absorption at a given dye), the extinction coefficient for the free dyes and the extinction coefficient for the dyes in the fully bound form, respectively.

#### 4.2.5. Molecular Model

Molecular model docking simulations were based on the AutoDock program (4.0, The Scripps Research Institute, San Diego, CA, USA) downloaded from the http://autodock.scripps.edu/) and DS Visualizer software (5.0, BIOVIA, San Diego, CA, USA). The three-dimensional structural information of BSA and DNA was from http://www.rcsb.org/pdb/ (ID: 3V03 for BSA and ID: 453D for CT-DNA, respectively). The 3D structures of the dyes were generated in Discovery Studio 4.5 Client. The grid center coordinates of the box along the x-, y-, z-axes were set to 126 Å, 126 Å and 126 Å for BSA–dye interactions and 88 Å, 86 Å and 126 Å for DNA–dye interactions, respectively. The Lamarckian genetic algorithm was employed to search the docking conformation.

## 5. Conclusions

In recent decades, whether temporary hair dyes do harm to human health has been controversial with limited data available to address this issue. For the first time, to the best of our knowledge, we have explored the bio-safety evaluation of three direct hair dyes (FB, BR2 and VBB), with ethically permitted strategies. In the present work, the percutaneous absorption, cytotoxic effect on RBC of human blood and NIH/3T3 cells, and the BSA/DNA interactions of these three hair dyes were comparatively studied. FB and BR2 can potentially be absorbed through human skin and all three dyes have cytotoxicity on human RBC and NIH/3T3 in very light concentration. Moreover, experimental and theoretical studies suggest that the three dyes have intense interactions with BSA and CT-DNA. This leads us to conclude that the three temporary hair dyes could result in significant potential risk to human health and those who use them should be aware of their potential toxic effects.

## Figures and Tables

**Figure 1 molecules-24-01744-f001:**
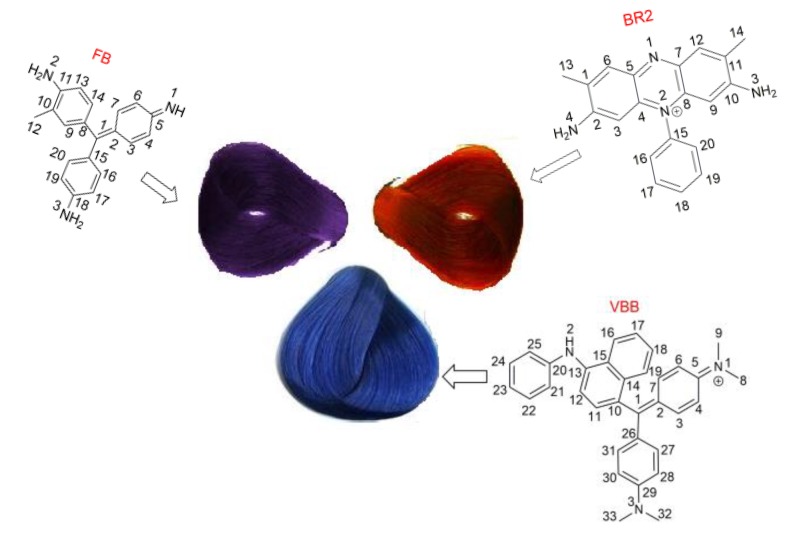
The molecular structures and decorated hair colors of hair dyes fuchsin basic (FB), basic red 2 (BR2) and Victoria blue B (VBB).

**Figure 2 molecules-24-01744-f002:**
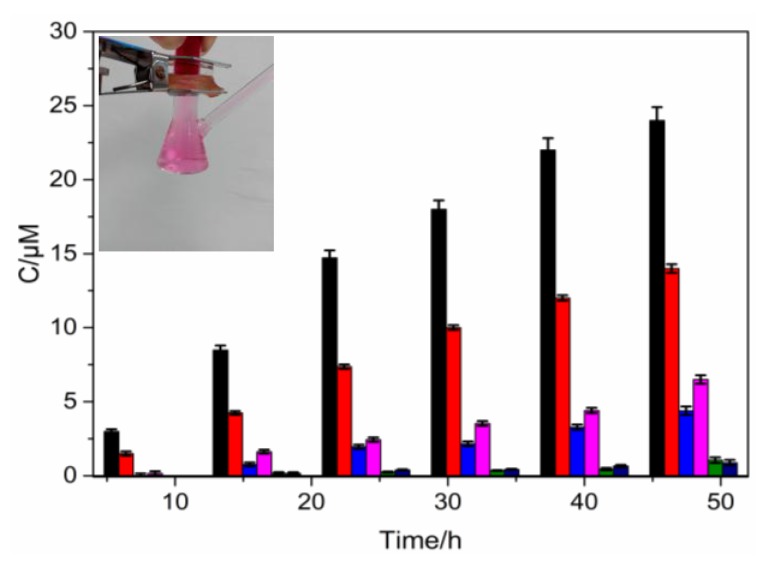
The percutaneous absorption of hair dyes. 0.3% (the largest amount without oxidants) and 0.15% (mixed with oxidants) for FB, 0.5% (the largest amount without oxidants) and 0.25% (mixed with oxidants) for VBB, and 0.4% (the largest amount without oxidants) and 0.2% (mixed with oxidants) for BR2, respectively. (*n* = 3).

**Figure 3 molecules-24-01744-f003:**
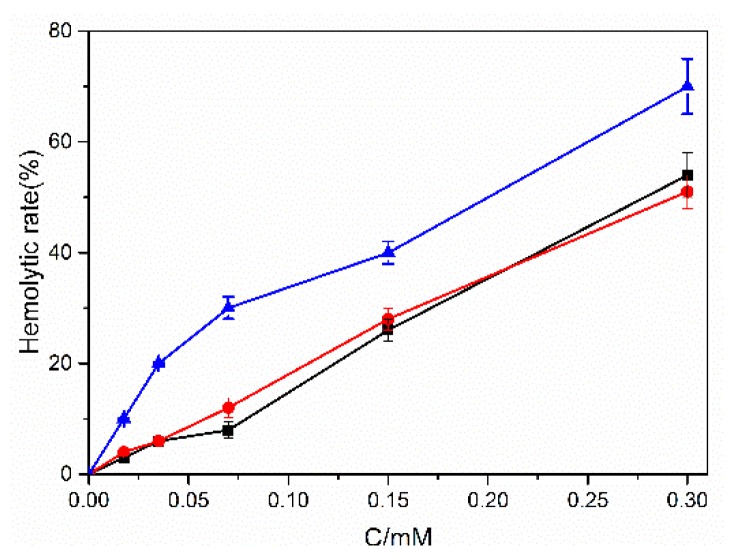
The hemolytic rate of human red blood cells (RBC) under different concentrations of hair dyes FB(●), BR2(■) and VBB(▲). (*n* = 3).

**Figure 4 molecules-24-01744-f004:**
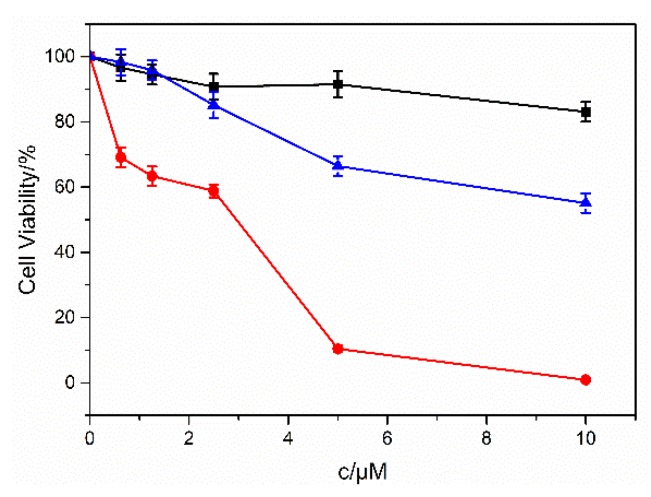
The viability change of NIH/3T3 cells when incubated with different concentrations of BR2 (■), FB (▲) and VBB (●) dyes. (*n* = 5).

**Figure 5 molecules-24-01744-f005:**
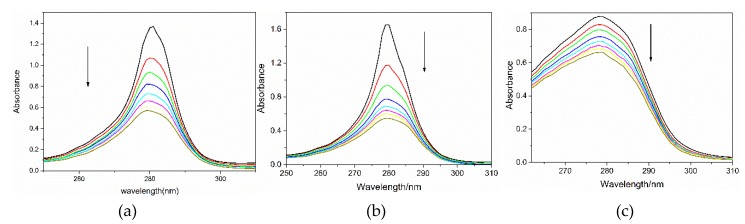
The absorption spectra of BSA under different concentrations of (**a**) BR2, (**b**) VBB and (**c**) FB. The arrows indicate the BSA spectra change with the addition of the dye molecules at different concentrations for 0 (**—**), 4(**—**), 8(**—**), 12(**—**), 16(**—**), 20(**—**), 24(**—**), and 28(**—**) μM. (*n* = 3).

**Figure 6 molecules-24-01744-f006:**
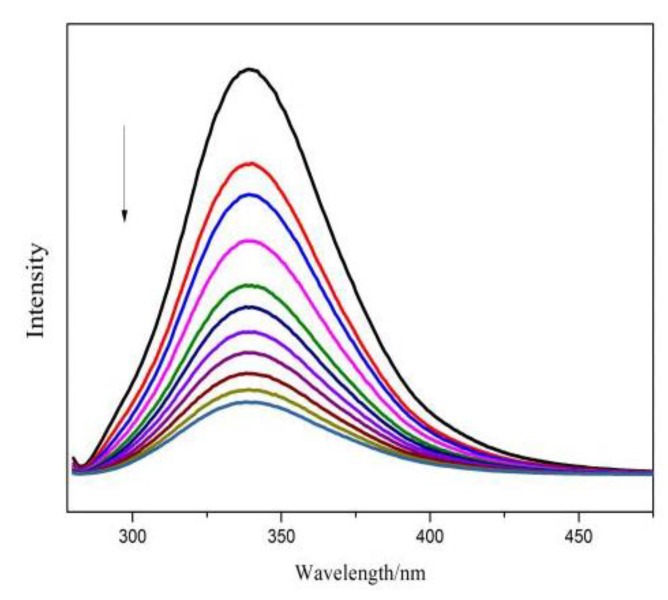
The emission spectra of BSA under different concentrations of FB. The arrows indicate the BSA spectra change with the addition of the dye molecules at different concentrations for 0 (**—**), 4(**—**), 8(**—**), 12(**—**), 16(**—**), 20(**—**), 24(**—**), 28(**—**), 32 (**—**), and 36(**—**) μM. (*n* = 3).

**Figure 7 molecules-24-01744-f007:**
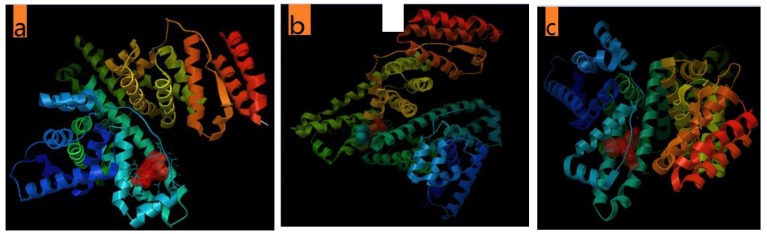
The most stable conformations between BSA and dyes (**a**) FB, (**b**) BR2 and (**c**) VBB based on the lowest free binding energy.

**Figure 8 molecules-24-01744-f008:**
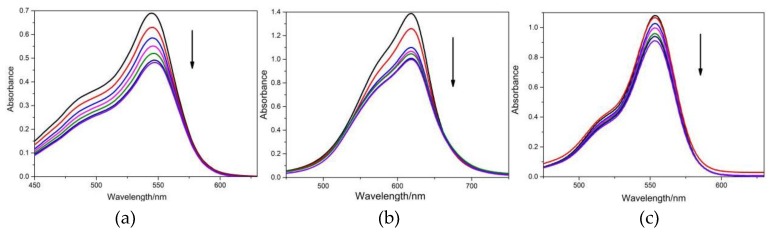
The absorption spectra of (**a**) FB, (**b**) BR2 and (**c**) VBB under different concentrations of CT-DNA. The arrows indicate the spectra change of dyes with the addition of CT-DNA at different concentrations for 0 (**—**), 4 (**—**), 8(**—**), 12 (**—**), 16 (**—**), 20 (**—**), and 24 (**—**) μM. (*n* = 3).

**Figure 9 molecules-24-01744-f009:**
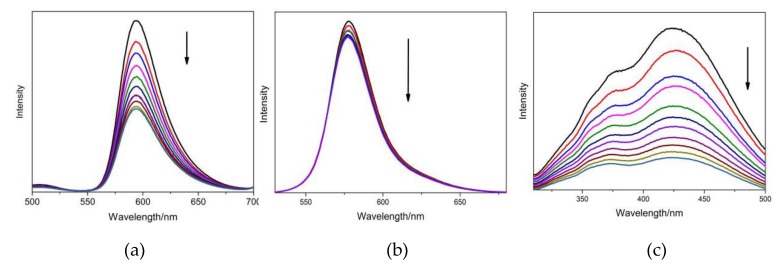
The emission spectra of (**a**) FB, (**b**) BR2 and (**c**) VBB under different concentrations of CT-DNA. The arrows indicate the spectra change of dyes with the addition of CT-DNA at different concentrations for 0 (**—**), 2 (**—**), 4 (**—**), 6 (**—**), 8 (**—**), 12 (**—**), 16 (**—**), 18 (**—**), and 20 (**—**) μM. (*n* = 3).

**Figure 10 molecules-24-01744-f010:**
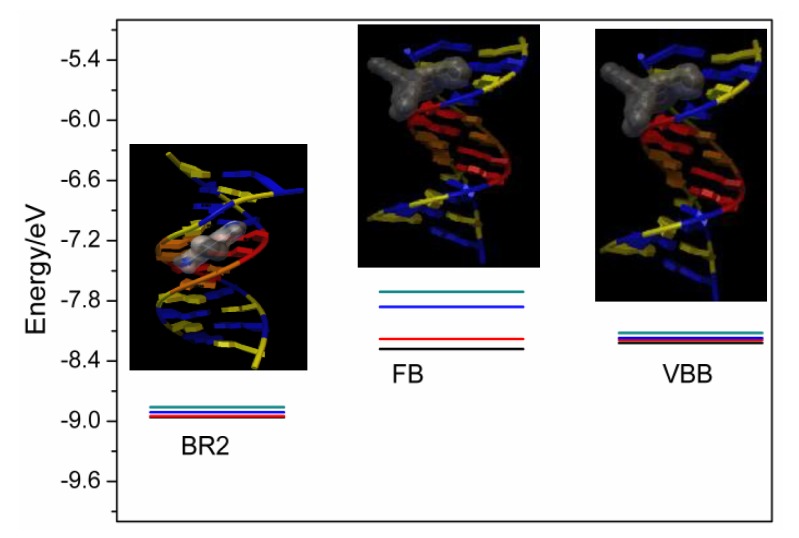
The lowest binding energies and the most stable conformations between FB, BR2 and VBB based on the lowest free binding energy.

**Table 1 molecules-24-01744-t001:** The lethal dose 50% (LD_50_) values on RBC and NIH/3T3 cells as well as the bovine serum albumin (BSA) and DNA-binding parameters of the hair dyes.

	^a^ LD_50_		BSA Binding		Calf Thymus (CT)-DNA Binding	
	RBC (mM)	NIH/3T3 (μM)	K_q_ × 10^1^^1^	K_b_ × 10^3^	*n*	K’_b_ × 10^2^
FB	0.23 ± 0.01	10.66 ± 0.31	23.4 ± 0.22	118 ± 1.82	1.21 ± 0.02	1.2 ± 0.04
BR2	0.29 ± 0.02	22.77 ± 0.43	1.1 ± 0.01	4.37 ± 0.01	0.88 ± 0.01	530 ± 3.35
VBB	0.16 ± 0.01	2.318 ± 0.03	21.9 ± 0.31	30.1 ± 0.54	1.03 ± 0.03	140 ± 2.48

^a^ LD_50_ values were evaluated after a defined period of treatment of each system to individual dyes. The results are mean values of three replicates. The values are statistically significant (*p* < 0.05) for each dye and system employed in this study.

**Table 2 molecules-24-01744-t002:** The hydrogen bonds of dyes with BSA and CT-DNA.

Dye	Atom	BSA	Distance (Å)	CT-DNA	Distance (Å)
FB	N1	His145 NE2	2.88	dg10 N2	3.21
FB	N2			dc9 O4	2.96
FB	N2			dt8 O2	3.25
BR2	N2	Lys499 N2	2.89		
BR2	N4	Glu125 OE2	2.91	dt19 O3	3.06
BR2	N3	Leu115 O	2.77		
BR2	N4	Gln416 OE	2.55		
VBB	N2			da6 O4	2.17

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
