# Peer review of "The Bio-Safety Concerns of Three Domestic Temporary Hair Dye Molecules: Fuchsin Basic, Victoria Blue B and Basic Red 2"

_molecules, 2019, doi:10.3390/molecules24091744_

Round 1

Reviewer 1 Report

This manuscript could be accepted. 

Author Response

Thank you for your help and kind instruction.

Reviewer 2 Report

Overall I find that the manuscript has improved significantly.

Some general concerns:

Figure legends, Could you please indicate experimental replicate numbers, n=x? and please specify error bars. 

Specific comments:

Figure 2: legend text could be more clear. On the curious note, are the concentrations with and without oxidizer? Moreover, are the investigated hair dyes use with oxidizer in consumer products?

Table1: You need to indicate the level of experimental variation.  

Figure 3. To me, E.coli toxicology is a bit off in this context and you do not conclude on these data. I suggest you take out this part and narrow down you overall aim. I think the manuscript will stand out more focused. 

Figure 4a; are the micrographs representative of the monolayer cultures? and is there a quantitative output to these depictions? Otherwise, I think you should remove this specific content. 

Figure 5 & 6: You need to help the reader and indicate the colors of the graphs, otherwise it the data stands out as uninterpretable. 

Author Response

Dear editor,     

Thank you and the reviewers very much for your time and thoughtful comments. These kind suggestions greatly improved the quality of our work and have been carefully considered in the revised manuscript. Below are our point-to-point responses to the Reviewer’s comments. For your convenience, the reviewers’ comments were given here in an italic font.

Reviewer 1:

1.      Figure legends, Could you please indicate experimental replicate numbers, n=x? and please specify error bars.

Response:

The experimental replicate numbers have been added for all the figure legends with colored text and the error bars have been clearly added in the figures.

2.      Figure 2: legend text could be more clear. On the curious note, are the concentrations with and without oxidizer? Moreover, are the investigated hair dyes use with oxidizer in consumer products?

Response:

    The legend of figure 2 has been revised to be clearer. The concentrations with or without oxidizers have been pointed out.

Moreover, in the consumer products, for permanent hair dyes, the researched dyes were always used with oxidizers such as H2O2, since the oxidizers can destroy the hair scales and help the dyes to enter the inner layer of the hair. In contrast, the temporary hair dyes were always used without the oxidizers and the largest concentrations were limited by FDA. Herein, we studied the percutaneous absorption of hair dyes for both the largest concentration without oxidizers and the concentration with concentration.

3.      Table1: You need to indicate the level of experimental variation.   

Response:

The level of experimental variation has been added in table 1.

4.      Figure 3. To me, E.coli toxicology is a bit off in this context and you do not conclude on these data. I suggest you take out this part and narrow down you overall aim. I think the manuscript will stand out more focused.

Response:

   We calculated the colony forming units of these dyes to E.coli to evaluate the damage of these hair dyes to the environment. Our results indicated that the commercially used dyes have low LD50 to E. coli, indicating that they are harmful to the environment when discharged to the environment.  

   Nevertheless, since our aim of this work is to research the biosafety of these hair dyes, we carefully considered the suggestion of the reviewer and agreed that taking out the E.Coli part could narrow down our aim. The E.coli toxicology study has been taken out from the context in the revised paper.

5.      Figure 4a; are the micrographs representative of the monolayer cultures? and is there a quantitative output to these depictions? Otherwise, I think you should remove this specific content. 

Response:

    Fig. 4a exhibited the observed results of monolayer cultures but regretfully we don’t have quantitative output to these depictions. We have removed this specific content from the revised paper according to the suggestion of the reviewer.

6.      Figure 5 & 6: You need to help the reader and indicate the colors of the graphs, otherwise it the data stands out as uninterpretable.

Response:

Thank you. The colors of the graphs have been indicated in the mentioned figures above.

Please don’t hesitate to contact us if you have other concerns. Thank you.

Sincerely yours,

Bing Liu and Ping Zhao

Round 2

Reviewer 2 Report

I accept the manuscript in its present form.

This manuscript is a resubmission of an earlier submission. The following is a list of the peer review reports and author responses from that submission.

Round 1

Reviewer 1 Report

Abstract is clear and seems sound. However, reading the remaining manuscript I am missing a clear aim of the study. From my perspective there need to be a hypothesis, which is falsified accordingly.

Throughout the experimental work, statistical considerations are not applied and moreover experimental variability is not indicated, nor is sample size.

There is no considerations of the proposed risk safety model on known hazardous chemicals like PPD

Reviewer 2 Report

In this article, Liu et.al. have addressed the environmental awareness of cosmetics specially the hair dyes in a very systemetic way. This work will be interesting to the scientific as well as industrial research. This article could be publishable.

However, I would like to highlight one point.  Few articles have been published (J. Haz. Mat., vol 338, 2017, page 356; Cosmetics, vol 2, 2015, page 313) recently. I think these are relevent in the context of this article. The authors can site these works if they want.

Thank you.